# Simultaneous Design of the Host Structure and the Polarisation Profile of Piezoelectric Sensors Applied to Cylindrical Shell Structures

David Ruiz [1], Sergio Horta Muñoz [2,*] and Reyes García-Contreras [2]

1 OMEVA Research Group, Escuela de Ingeniería Industrial y Aeroespacial de Toledo, Universidad de Castilla-La Mancha, Av. Carlos III, Campus Fábrica de Armas, 45004 Toledo, Spain; david.ruiz@uclm.es

2 Instituto de Investigación Aplicada a la Industria Aeronáutica, Escuela de Ingeniería Industrial y Aeroespacial de Toledo, Universidad de Castilla-La Mancha, Av. Carlos III, Campus Fábrica de Armas, 45004 Toledo, Spain; mariareyes.garcia@uclm.es

* Correspondence: sergio.horta@uclm.es

**Abstract:** Piezoelectric actuators and sensors are applied in many fields in order to produce forces or displacements with the aim of sensing, manipulating or measurement, among other functions. This study presents the numerical methodology to optimize the static response of a thick-shell structure consisting of piezoelectric sensors, based on the maximisation of the electric charge while controlling the amount of piezoelectric and material required. Two characteristic functions are involved, determining the topology of the sensor and the polarisation profile. Constraints over the reaction force are included in the optimisation problem in order to avoid singularities. The topology optimisation method is used to obtain the optimal results, where regularisation techniques (density filtering and projection) are used to avoid hinges. The minimum length scale can be controlled by the use of three different projections. As the main novelty, a displacement-controlled scheme is proposed in order to generate a robust algorithm for future studies including non-linearities.

**Keywords:** topology optimisation; piezoelectric actuator; shell; finite element method

**MSC:** 74P15

## 1. Introduction

The topology optimisation method is a conceptual tool which allows us to increase the capabilities of different types of devices. The classical mechanical problem is the minimisation of the compliance or the weight of a structure, but in the last years this method has been used in different fields of science such as electronics, propagation of waves and optics, among others. This paper is focused on the improvement of the response of piezoelectric sensors, with the objective of reducing the size of the device to increase the range of applications.

The application of piezoelectric sensors and actuators has experienced significant development in recent years. Piezoelectric sensors are devices that produce a small voltage when they are deformed, while piezoelectric actuators take advantage of the ability to generate a displacement when voltage is applied. This effect is generally used in situations that require the application of large forces in an ultra-precise way [1], as well as to generate systems capable of developing handling functions at a microscopic level [2–4].

One of the common applications is the placement of piezoelectric patches on structures subjected to vibrations, so that it is possible to monitor the state of vibrational states and control undesirable vibrations, generating so-called smart structures [5–11]. In these structures, the location of the piezoelectric elements is critical, due to the need to adjust the positioning so that their effect is maximised, reducing the cost of the material to be used. In addition, another critical design factor in structural elements is usually weight, so it is of

interest to minimize the volume of material used in the host structure, while maintaining certain levels of rigidity [12,13]. Another application associated with the maximisation of electrical load obtained by piezoelectric sensors is energy harvesting systems. This has also significantly grown in recent years [14] considering that they can be used as a way for recovering waste energy from many surfaces, not only in industry, but also in our daily lives. A particular example of their applicability is the shock absorbers of vehicles which are cylindrical elements that are subjected to vibration loads during service [15].

In addition, the use of cylindrical shells is increasingly widespread in engineering applications, mainly in sectors such as civil, chemical, aerospace and naval transportation [8]. In these sectors, the structural analysis of shell elements such as pressurised tanks, aircraft fuselage, fuel tanks or fluid pipes are commonly found in literature [16,17]. These structures have characteristics such as high rigidity and lightweight, which leads to their application in loading conditions resulting in high level of stresses. It is interesting to use piezoelectric elements for the purpose of Structural Health Monitoring (SHM), or to modify the shape of the structure to improve its structural response [18,19] or aerodynamics [20]. There are numerous works that seek to analyse the response of shell-type structures with piezoelectric layers from the analytical, numerical and experimental points of view [21–23]. For instance, Yue et al. [9] experimentally measured the capacity for sensing and vibration control with piezoelectric patches in a paraboloidal shell structure, which can be implemented in structonic systems typical of aerospace sector. Similarly, Li et al. [8] theoretically estimated and experimentally validated the effect of the orientation of diagonal piezoelectric sensors in a cylindrical shell excited by piezoelectric actuators. It is also worth mentioning in this field the work of Varelis and Saravanos [24], in which the ability to predict the non-linear electromechanical response of laminated piezoelectric shell under buckling and elastic instability is analytically demonstrated. In this work, a commonly used iterative technique is maintained, i.e., Newton–Raphson, therefore the Cylindrical Arc-Length method was applied in order to overcome the snap-through points.

Previous works [13,25,26] have shown that the implementation of topological optimisation on numerical models based on the Finite Element Method (FEM) allows, simultaneously, an optimizing host structure and a polarisation profile of the electrodes. These works were carried out on different geometries in the form of flat plates and one-dimensional beams. A similar work, applied in this case to curved shell-type structures, was carried out by Donoso et al. [27], but limited to the design of the polarisation profile.

Nevertheless, none of the previously mentioned studies apply topological optimisation to the simultaneous design of the support structure and the polarisation profile in shell elements. The present work develops the numerical modelling that allows this optimal design, maximizing the electric energy produced and allowing the application of restrictions on the volume of material, in order to achieve a light and low-cost structure. In addition, regularisation techniques [28–30] are used in order to avoid the appearance of hinges. Unlike previous works by the authors [13,25,26], a control scheme based on the application of displacement was specifically developed, in contrast to the usual approach of the compliance optimisation problem which takes the applied force as a reference. This control scheme may avoid a lack of convergence when snap-through issues arise [31,32].

The work is divided as follows. Section 2 describes the mathematical formulation of the electric charge and the mechanical elastic response of the shell. The mathematical formulation of the optimisation problem is presented in Section 3. Numerical results are found in Section 4. Finally, the conclusions of the work are shown in Section 5.

## 2. Formulation of the Problem

### 2.1. Governing Equations

The computation of the electric charge $q$, which represents the capacity of the piezoelectric sensor, is obtained following Equation (1) [33]. This equation is simplified considering the negligible effect of the piezoelectric layer on the stiffness of the structure and the piezoelectric isotropy ($e_{31} = e_{32}$) of the sensor [27].

$$q = e_{31} \int_{\Omega} \chi_p(x_1, x_2)[\varepsilon_{11} + \varepsilon_{22}] \mathrm{d}\Omega = e_{31} \int_{\Omega} \chi_p(x_1, x_2) \left[ \frac{\partial u}{\partial x_1} + \frac{\partial v}{\partial x_2} + x_3 \left( \frac{\partial \phi_2}{\partial x_1} - \frac{\partial \phi_1}{\partial x_2} \right) \right] \mathrm{d}\Omega, \tag{1}$$

where $(u, v)$ are the translational in-plane displacements, $(\phi_1, \phi_2)$ the rotation over the $x_1$ and $x_2$-axis, respectively, $\Omega$ is the design domain and $e_{31}$ is the piezoelectric constant, i.e., a material property. $\chi_p \in \{-1, 0, 1\}$ is a characteristic function that represents the polarity of the surface electrode, $\varepsilon_{11}$ and $\varepsilon_{22}$ are the in-plane normal strains.

The displacements and rotations are calculated by solving the equilibrium equation:

$$\begin{cases} -\mathrm{div}(\mathbf{E_s}(\chi_s) : \boldsymbol{\varepsilon}) &= f_v, & \text{in} & \Omega \\ (\mathbf{E_s}(\chi_s) : \boldsymbol{\varepsilon}) \cdot \mathbf{n} &= f_s, & \text{in} & \Gamma_f \end{cases},$$

subject to the boundary conditions:

$$\begin{cases} u, v, w &= 0, & \text{in} & \Gamma_c \\ u &= u_{in} & \text{in} & \Gamma_u \\ v &= v_{in} & \text{in} & \Gamma_v \\ w &= w_{in} & \text{in} & \Gamma_w \end{cases},$$

with $w$ the vertical displacement, $\mathbf{E_s}$ the stiffness tensor, $\boldsymbol{\varepsilon}$ the infinitesimal strain tensor, $f_v$ and $f_s$ the volumetric and surface forces, respectively. $\Gamma_f$ and $\Gamma_c$ represent the boundary of $\Omega$ where forces are imposed and displacements are constraint, respectively, $\mathbf{n}$ the normal vector of the boundary and $u_{in}$, $v_{in}$ and $w_{in}$ the displacements imposed in $\Gamma_u$, $\Gamma_v$ and $\Gamma_w$. $\chi_s \in \{0, 1\}$ represents the host structural variable that defines void or solid, respectively.

### 2.2. Finite Element Model

Flat thick-shell formulation is developed based on Reissner-Mindlin plate theory for a bidimensional finite element consisting of four nodes with six degrees of freedom (DOF), three displacements $u$, $v$ and $w$, and three rotations $\phi_1$, $\phi_2$ and $\phi_3$ [34,35], described with regard to an element local coordinate system $(x_1, x_2, x_3)$. Displacement and rotations are defined independently and therefore they are interpolated separately. The interpolation of in-plane displacements, associated to membrane behaviour, is shown in Equation (2).

$$\begin{bmatrix} \tilde{u} \\ \tilde{v} \end{bmatrix} = \mathbf{N_m} \begin{bmatrix} u_i \\ v_i \end{bmatrix}, \tag{2}$$

where $\tilde{u}$, $\tilde{v}$ are the element interpolated displacements, $\mathbf{N_m}$ is the shape functions matrix for a quadrinodal membrane element and subscript $i \in \{1, 2, 3, 4\}$ refers to the specific node. The bending DOFs, representing the out-of-plane displacements and rotations, are interpolated applying bending shape functions as shown in Equation (3).

$$\begin{bmatrix} \tilde{w} \\ \tilde{\phi}_1 \\ \tilde{\phi}_2 \end{bmatrix} = \mathbf{N_b} \begin{bmatrix} w_i \\ \phi_{1i} \\ \phi_{2i} \end{bmatrix}, \tag{3}$$

with $\mathbf{N_b}$ being the bending shape functions matrix.

The stiffness matrix in local element coordinates is obtained by concatenating the membrane matrix (defined in Equation (4)), corresponding to the two in-plane translational displacements ($u$ and $v$), while the bending terms (Equation (5)) are obtained from the thick-plane element, which consists of three DOFs ($w$, $\phi_1$ and $\phi_2$). The sixth DOF, $\phi_3$, is assigned an arbitrary stiffness, much lower than the rest of components, taking into consideration that this rotation does not contribute to strain energy [34]. Nevertheless, this DOF is required for consistency of matrices when transforming to the global coordinate system. The integration in the domain of the element ($\Omega^e$) is reduced to an integration in the

area ($A$), described in the $x_1$ and $x_2$ directions. This integration is performed numerically using a reduced integration scheme based on Gaussian Quadrature to avoid shear locking.

$$\mathbf{K_m} = \int_{\Omega^e} \mathbf{B_m}^T \mathbf{C_m} \mathbf{B_m} \, d\Omega^e = \int_A \left( \int_0^h dx_3 \right) \mathbf{B_m}^T \mathbf{C_m} \mathbf{B_m} \, dA = h \int_A \mathbf{B_m}^T \mathbf{C_m} \mathbf{B_m} \, dA \quad (4)$$

$$\mathbf{K_b} = \int_A \mathbf{B}_b^T \frac{h^3}{12} \mathbf{C}_b \mathbf{B}_b \, dA + \int_A \mathbf{B}_s^T hk \mathbf{C}_s \mathbf{B}_s \, dA, \quad (5)$$

where $\mathbf{B}$ is the derivative of the shape functions, $\mathbf{C}$ the material stiffness tensor, particularised in this study for a linear isotropic elastic material and $h$ is the thickness of the element (dimension in $x_3$-direction). The subscripts $b$, $m$ and $s$ represent bending, membrane and shear, respectively. Finally, $k$ represents the stiffness associated with the drilling DOF ($\phi_3$), the value of which is about one-thousandth of the smallest diagonal element of the element matrix stiffness, following recommendations in the literature [34]. More information about the definition of these parameters could be found in finite element reference books [34,35].

Additionally, with the aim of computing the electric charge generated by the piezoelectric elements, it is necessary to compute the sum of strains in each element. This is defined in local coordinates in Equation (6), which can be related to the discretised problem by means of the derivative of shape functions.

$$\begin{bmatrix} \tilde{\varepsilon}_{11} \\ \tilde{\varepsilon}_{22} \\ \tilde{\varepsilon}_{12} \end{bmatrix} = \begin{bmatrix} \dfrac{\partial \tilde{u}}{\partial x_1} \\ \dfrac{\partial \tilde{v}}{\partial x_2} \\ \dfrac{\partial \tilde{u}}{\partial x_2} + \dfrac{\partial \tilde{v}}{\partial x_1} \end{bmatrix} - x_3 \begin{bmatrix} -\dfrac{\partial \tilde{\phi}_2}{\partial x_1} \\ \dfrac{\partial \tilde{\phi}_1}{\partial x_2} \\ \dfrac{\partial \tilde{\phi}_1}{\partial x_1} - \dfrac{\partial \tilde{\phi}_2}{\partial x_2} \end{bmatrix} = \mathbf{B_m} \begin{bmatrix} u_i \\ v_i \end{bmatrix} - x_3 \mathbf{B_b} \begin{bmatrix} w_i \\ \phi_{1i} \\ \phi_{2i} \end{bmatrix}. \quad (6)$$

As the geometry to be modelled is not coplanar, the elements have different local orientations, therefore it is necessary to compute the global stiffness matrix in global coordinates, which are called $xyz$. The rotation could be performed by means of a transformation matrix defined by the direction cosines relating to both coordinate systems.

## 3. Topology Optimisation Problem and Sensitivity Analysis

In this work we aim to maximize the electric charge produced in a cylindrical-type structure submitted to a static deformation. The expression for the discretised objective function is:

$$q = \mathbf{F}^T(\boldsymbol{\rho}_p, \boldsymbol{\rho}_s)\mathbf{U} = \sum_e^{n_{el}} \rho_{pe} \rho_{se}^3 \mathbf{B}_e^T \mathbf{U}_e, \quad (7)$$

where $n_{el}$ is the number of finite elements, $\mathbf{B}_e$ is the discretisation of the strain displacement matrix, $\mathbf{U}_e$ is the vector with the displacement of the element $e$. The variable $\rho_{pe}$ defines the sign of the polarisation profile, while the role of the relaxed variable $\rho_{se}$ is to penalize the electric charge generated by void elements [36]. The piezoelectric property $e_{31}$ has been removed from the objective function, since a constant does not affect the optimal design. The constraint over the maximum volume fraction is included in the problem, as this usually improves the convergence of the optimisation algorithm. The global stiffness of the structure is controlled by adding two constraints over the reaction forces in the structure. This ensures that the point where the displacement is imposed is connected with the boundary conditions. Finally, taking into account Equation (7), the formulation of the discretised problem is stated as follows:

$$\max_{\boldsymbol{\rho}_s, \boldsymbol{\rho}_p} : \quad q$$

subject to:

$$
\begin{cases}
\tilde{\boldsymbol{\rho}}_s &= \mathbf{H}(\boldsymbol{\rho}_s) \\
\hat{\boldsymbol{\rho}}_s &= \mathbf{P}(\tilde{\boldsymbol{\rho}}_s) \\
\mathbf{K}(\hat{\boldsymbol{\rho}}_s)\mathbf{U} &= \mathbf{R} \\
\mathbf{L}_u^T\mathbf{U} &= u_{in} \\
\mathbf{v}^T(\hat{\boldsymbol{\rho}}_s) &\leq V_0\,|\,\Omega\,| \\
\mathbf{L}_r^T\mathbf{R} &\leq r_{max} \\
\mathbf{L}_r^T\mathbf{R} &\geq r_{min}
\end{cases} ,
$$

where $\mathbf{L}_r$ is a vector of zeros with the value 1 in the constrained degrees of freedom, $\mathbf{R}$ is the reaction force vector, $\tilde{\boldsymbol{\rho}}_s$ is the filtered structural density, $\hat{\boldsymbol{\rho}}_s$ is the projected density [37], $\mathbf{L}_u$ is a vector of zeros with the value 1 in the degree of freedom where the displacement is imposed, $u_{in}$ is the fixed displacement, $\mathbf{v}$ is a vector containing the measure of the elements, $V_0$ is the maximum volume fraction, $|\,\Omega\,|$ is the measure of the design domain, finally, $r_{max}$ and $r_{min}$ are the maximum and minimum reaction force allowed, respectively, used to avoid singular solutions.

The well-known Solid Isotropic Material with Penalisation (SIMP) method [28] is used to penalize intermediate densities. The expression for a smoothed threshold projection [29] based on the hyperbolic tangent function is:

$$
\hat{\rho}_{se} = \frac{\tanh(\beta\eta) + \tanh(\beta(\tilde{\rho}_{se} - \eta))}{\tanh(\beta\eta) + \tanh(\beta(1 - \eta))}, \tag{8}
$$

where $\eta \in [0,1]$ and $\beta$ are tuning parameters that define the threshold and the sharpness of the function, respectively. The filtered densities of Equation (8) are projected to 0 or 1 depending if these value are smaller or bigger than the threshold $\eta$. The filtered densities $\tilde{\boldsymbol{\rho}}$ are expressed as [30]:

$$
\tilde{\rho}_{se} = \frac{\displaystyle\sum_{j}^{n_{el}} d_e(\mathbf{x}_j)\rho_{sj}}{\displaystyle\sum_{j}^{n_{el}} d_e(\mathbf{x}_j)},
$$

where $\mathbf{x}_j$ is the barycentre of the $j$-th element, and the weighting function $d_e(\mathbf{x}_j)$ is given by the cone-shape function:

$$
d_e(\mathbf{x}_j) = \max\{R_f - ||\mathbf{x}_j - \mathbf{x}_e||, 0\},
$$

where $R_f$ is the filter radius.

The use of the filtering technique together with the projection method ensures a mesh-independent 0–1 design. As shown in [25], the polarisation variable $\boldsymbol{\rho}_p$ does not need any kind of regularisation.

### 3.1. Robust Formulation

This section presents the robust formulation of the problem, which was introduced in [29]. This consists of the use of three different projections called erode, intermediate and dilate and from now on, the projection will be represented with the superscript $(m)$ for each projection $((e)$, $(i)$ and $(d)$, respectively). The implementation of this approach ensures a minimum length scale in both void and solid regions, hence avoiding the appearance of hinges.

The robust topology optimisation problem is written in terms of a min-max problem, which is not differentiable. The problem is then reformulated using the so-called bound formulation:

$$
\max_{\boldsymbol{\rho}_s, \boldsymbol{\rho}_p} : \quad \alpha \tag{9}
$$

subject to:

$$
\begin{cases}
q^{(m)} & \geq & \alpha \\
\tilde{\boldsymbol{\rho}}_s & = & \mathbf{H}(\boldsymbol{\rho}_s) \\
\hat{\boldsymbol{\rho}}_s^{(m)} & = & \mathbf{P}^{(m)}(\tilde{\boldsymbol{\rho}}_s) \\
\mathbf{K}(\hat{\boldsymbol{\rho}}_s^{(m)})\mathbf{U}^{(m)} & = & \mathbf{R}^{(m)} \\
\mathbf{L}_u^T\mathbf{U}^{(m)} & = & u_{in} \\
\mathbf{v}^T\hat{\boldsymbol{\rho}}_s^{(d)} & \leq & V_0^* \mid \Omega \mid \\
\mathbf{L}_r^T\mathbf{R}^{(m)} & \leq & r_{max} \\
\mathbf{L}_r^T\mathbf{R}^{(m)} & \geq & r_{min} \\
m & \equiv & \{e, i, d\},
\end{cases}
\tag{10}
$$

where $\alpha$ is an additional bound variable, superscript $(m)$ represents the projection and $V_0^* = \dfrac{V_0}{V^{(i)}} V^{(d)}$ is the maximum volume fraction allowed for the dilate projection. This value is updated every 20 iterations. This formulation solves the non-differentiability issue with the max–min function. It is important to remark that the equilibrium equation and the constraints of the reaction forces must be computed for each projection.

### 3.2. Computation of Sensitivities

The optimisation problem is solved using the Method of the Moving Asymptotes (MMA) [38]. This algorithm needs the partial derivatives with respect to the variables $\boldsymbol{\rho}_s$ and $\boldsymbol{\rho}_p$.

The derivatives of the elastic problem equations (the equilibrium equations and the constraints) are straightforward, and they are not included in this work for the sake of brevity. The derivative of the function $q$ with respect to $\boldsymbol{\rho}_s$ is computed using the chain rule:

$$
\frac{\partial q}{\partial \rho_{se}} = \frac{\partial q}{\partial \hat{\rho}_{se}} \frac{\partial \hat{\rho}_{se}}{\partial \tilde{\rho}_{se}} \frac{\partial \tilde{\rho}_{se}}{\partial \rho_{se}},
$$

with:

$$
\frac{\partial q}{\partial \hat{\rho}_{se}} = \left( \frac{\partial \mathbf{F}^T}{\partial \hat{\rho}_{se}} \mathbf{U} + \mathbf{F}^T \frac{\partial \mathbf{U}}{\partial \hat{\rho}_{se}} \right).
$$

Note that the adjoint method can be used to circumvent the computational cost of computing the derivative of the displacement vector $\mathbf{U}$. The derivatives of $q$ with respect to $\boldsymbol{\rho}_p$ is:

$$
\frac{\partial q}{\partial \rho_{pe}} = \frac{\partial \mathbf{F}^T}{\partial \rho_{pe}} \mathbf{U}.
$$

In practice, it is convenient to work with normalised parameters in order to avoid computations with numbers with different magnitude order. The electrical charge is normalised with the electrical charge generated by the homogeneous design.

A summary of the process is shown in Algorithm 1.

---

**Algorithm 1:** Algorithm and computational implementation

---

    **Set**       : material properties, geometry and BC's
    **Set**       : Optimisation parameters
    **Define**  : initialisation $\rho_p$ and $\rho_s$
    **Compute**: reference charge $q_{ref}$
    **Set**       : Optimisation method tolerance *tol*
    **While** $e > tol$
        |  **Filtering and projection** $\rho_s \rightarrow \tilde{\rho}_s \rightarrow \hat{\rho}_s$;
        |  **Assembly** of global matrix $\mathbf{K}(\hat{\rho}_s)$ and vector $\mathbf{F}(\hat{\rho}_s, \rho_p)$;
        |  **Get** vector $\mathbf{U}$;
        |  **Compute** objective function $c = q$;
        |  **Compute** constraints;
        |  **Calculate** derivatives;
        |  **Update** variables with MMA $(\rho_s^*, \rho_p^*)$;
        |  **Define convergence variable** $e = ||(\rho_s^*, \rho_p^*) - (\rho_s, \rho_p)||$;
    **end**

---

## 4. Numerical Examples

Commercial software Matlab R2020b has been used to solve the finite element models and the optimisation problem proposed in this work. The results obtained, in terms of force, displacement, stress and strain fields, have been validated by means of the comparison with a commercial FEM software, i.e., Abaqus 2019 [39].

### 4.1. First Example

The domain $\Omega$ is defined as a semicylindrical shell. The dimensions are $L_x = 1$ m and $L_y = 1$ m with a global thickness of $t = 0.01$ m. The Young's modulus of the material is set to $E = 1$ Pa and the Poisson's ratio to $\nu = 0.3$.

The proposed structure is discretised in $60 \times 60$ elements. The scheme of the structure and its boundary conditions are shown in Figure 1. The displacements and rotations over the red lines are fixed to zero (clamped), while vertical displacement is imposed at the coordinates $(x, y, z) = (0, 0.5, 0.5)$ m with a value of $u_{in} = 0.15$ mm.

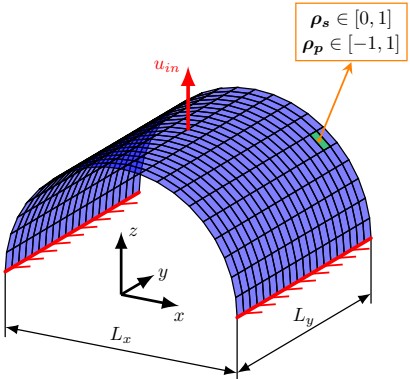

**Figure 1.** Dimensions and boundary conditions.

This case study is focused in obtaining the optimal electrode profile $\rho_p$ that will be used as initialisation in the rest of the examples. Since the host structure $\rho_s$ is fixed, it makes no sense to add constraints over the reaction force.

The result of the optimisation process is shown in Figure 2. The structure variable $\rho_s$ is represented in Figure 2 (left), with the black colour showing solid areas. The electrode profile appears at the centre, where blue and pink mean electrodes of different polarity. The whole structure including electrodes is depicted in Figure 2 (right).

  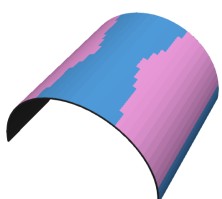

**Figure 2.** Structure layout $\rho_s$ (**left**), electrode profile $\rho_p$ (**centre**) and 3D design (**right**) for the first example.

The reference value used to compare the results is the cost generated by the homogeneous design $\rho_s = 1$ and $\rho_p = 1$. The cost excluding the piezoelectric constant $e_{31}$, is $\frac{c_{ref}}{e_{31}} = 0.721$ m$^2$. For the rest of examples, the objective function is the non-dimensional parameter defined as: $\lambda = \frac{c}{c_{ref}}$.

The value of the objective function for this first example is $\lambda_1 = 31.92$, showing the importance of the optimisation process. This value is larger than the reference, since the homogeneous electrode $\rho_p = 1$ is far from being a good design. The polarisation profile in Figure 2 shows that approximately half of the surface shell is subjected to strain with the opposite sign, and then most of the electrical charge produced by the positive polarity is cancelled with charge generated by the negative electrode.

The result of the optimisation process shows that the electrode profile obtained for each finite element is related to its curvature. This example clearly demonstrates that the optimisation of only one variable, the electrode—polarisation $\rho_p$—increases the electric charge generated by the sensor.

*4.2. Second Example*

The volume fraction is fixed to $V_0 = 0.5$ and the reaction force to $r = -3 \times 10^{-9}$ N. The values of $r_{min}$ and $r_{max}$ are computed by subtracting and adding a small value $\epsilon = r/100$. Concerning the tuning parameters of the filter and the projections, the filter radius is set to $R_f = 0.1$ m, the smoothness of the projection to $\beta = 1$ at the beginning of the iterative process, and it doubles the value every 40 iterations up to $\beta = 8$. The thresholds for the three projections are $\eta_e = 0.7$, $\eta_i = 0.5$ and $\eta_d = 0.3$, for the erode, intermediate and dilate projection, respectively.

The variable $\rho_s$ is initialised with a homogeneous design according to the volume constraint, and $\rho_p$ with the optimised polarity profile of the previous example. The optimal design is shown in Figure 3 (right).

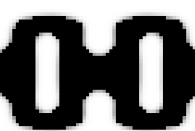 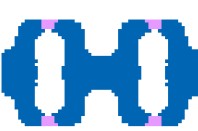 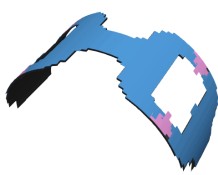

**Figure 3.** Structure layout $\rho_s$ (**left**), electrode profile $\rho_p$ (**centre**) and 3D design (**right**) for the second example.

The value of the objective function for the optimum design is $\lambda_2 = 20.91$. This result surpasses the reference charge, however, this value is smaller than $\lambda_1$. This is due to the maximum volume fraction imposed. The smaller the volume fraction is, the bigger the displacements are since the structure is less stiff, but the region $\Omega$ is also smaller. It is very convenient to use this constraint as this improves the convergence of the topology optimisation problem, as well as this can be used to control the amount of material if we have in mind the fabrication cost, the weight or the size of the structure.

The structural variable $\rho_s$ depicted in Figure 3 (left) shows that the whole structure is continuous, in the way that the point of application of the mechanical force is connected with the clamped edges. The robust scheme is working properly, which is corroborated by the absence of hinges.

### 4.3. Third Example

For this case study, the value of constraint over the reaction force is fixed to $r = -4 \times 10^{-9}$ N, while the rest of parameters do not change. This variation of the reaction force increases the structure stiffness, since the imposed displacement is the same as in the previous example. The results are shown in Figure 4.

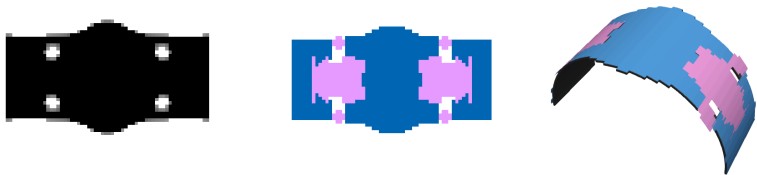

**Figure 4.** Structure layout $\rho_s$ (**left**), electrode profile $\rho_p$ (**centre**) and 3D design (**right**) for the third example.

The value of the objective function for the optimum design is $\lambda_3 = 60.44$. With this method, the stiffness of the structure can be modelled by imposing a different constraint $r$. This parameter can be adapted to the function of the application, since this is part of the input data.

In this last example the structure layout $\rho_s$ is stiffer than in the previous case. This is due to the reaction force, which is 25% higher than in the second example. This parameter can be fixed depending on the proposed application of the sensor.

### 4.4. Validation of the Results

The finite element problem has been solved by using an ad hoc script developed with the software Matlab. In order to validate the results obtained, the displacement field (the control variable in the optimisation problem) has been checked with Abaqus in the reference design (first example).

For the reference example, the deformed structure is shown in Figure 5, where the displacement has been scaled in order to better observe the deformed structure. It can be visually verified that the deformation obtained with both softwares is similar.

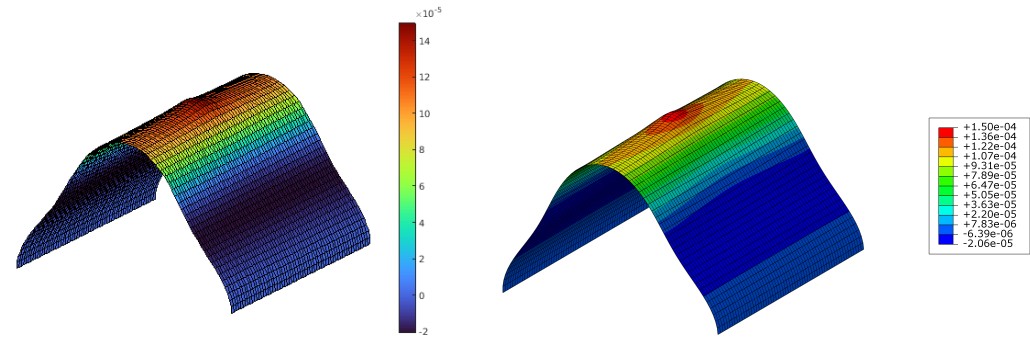

**Figure 5.** Deformation of the structure obtained with Matlab (**left**) and Abaqus (**right**).

Additionally, to corroborate that the finite element method has been correctly implemented, the vertical displacement of the midline (arc with coordinate $y = L_y/2$ following Figure 1) of the sensor is compared. To avoid a high relative error, the infinity norm has been used to compare the difference between both softwares:

$$||\tilde{\mathbf{w}}_M - \tilde{\mathbf{w}}_A||_\infty = 2.3740 \times 10^{-6} \text{ m},$$

where $\tilde{\mathbf{w}}$ represents the vertical displacement computed at the midline, and subscripts $M$ and $A$ stand for Matlab and Abaqus, respectively.

The vertical reaction forces computed at the node where the displacement is imposed are $\tilde{r}_M = 7.477 \times 10^{-9}$ N and $\tilde{r}_A = 7.551 \times 10^{-9}$ N. With a difference of $\approx 1\%$ we can consider the results obtained with Matlab valid.

## 5. Conclusions

In this work, a systematic procedure to maximize the electric charge generated by a semi-cylindrical piezoelectric sensor is presented. The objective function is computed in terms of two variables related through the deformation of the structure, the topology of the sensor and the polarisation profile of the electrode. The main novelty presented in this paper is the simultaneous optimisation of both variables.

The advantage of solving an optimisation problem is shown in several optimal designs, showing that the electric charge of the device has been improved for different volume fractions and values of the reaction force. The well-known issue of the appearance of hinges is overcome by implementing a robust scheme with three different projections. This regularisation also allows us to control the minimum length scale.

The shell modelled in this work is subjected to small displacements and small strains, but a control scheme based on the application of displacement (instead of controlling the applied force) is implemented with the objective of modelling a geometrically non-linear problem in the future.

In order to validate the mechanical response of the structure, the displacement field of the shell is computed with two different commercial softwares—Matlab and Abaqus.

**Author Contributions:** Conceptualisation, D.R.; investigation, D.R. and S.H.M.; writing—original draft preparation, D.R. and S.H.M.; writing—review and editing, D.R., S.H.M. and R.G.-C.; supervision, R.G.-C.; funding acquisition, D.R., S.H.M. and R.G.-C. All authors have read and agreed to the published version of the manuscript.

**Funding:** This project has been funded through grant PID2020-116207GB-I00 from the Spanish Ministerio de Ciencia e Innovación and SBPLY/19/180501/000110 from Junta de Castilla-La Mancha. In addition, the authors also acknowledge the financial support provided by the University of Castilla-La Mancha and the ERDF under the grants 2018/11744 and 2020/3771.

**Institutional Review Board Statement:** Not applicable.

**Informed Consent Statement:** Not applicable.

**Data Availability Statement:** Not applicable.

**Conflicts of Interest:** The authors declare no conflict of interest.

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
