# Peer review of "Simultaneous Design of the Host Structure and the Polarisation Profile of Piezoelectric Sensors Applied to Cylindrical Shell Structures"

_mathematics, doi:10.3390/math10152753_

Round 1
Reviewer 1 Report
In the present manuscript, the authors have developed a numerical methodology to optimize the static response of a thick-shell structure consisting of piezoelectric sensors, based on the maximization of the electric charge while controlling the amount of piezoelectric and material required. The article's overall presentation style and contents are fine; however, the authors must address and incorporate the following comments to publish it in Mathematics Journal.
1. The authors have carried out the topology optimization in this study. The author needs to address the motivation for using a topology optimization-based approach for thick-shell structures consisting of piezoelectric sensors.
2. In section 4.1 first example, the name of the used material for the semicylindrical shell is missing, the authors need to add the name and reference of the material properties used in the manuscript.
3. The authors have mentioned that the proposed model results in terms of force, displacement, stress, and strain fields, have been validated by means of the comparison with a commercial FEM software, i.e., Abaqus 2019. However, there is no such comparison found in the manuscript. The authors should make a table and compare the current model results with Abaqus FE results.
4. The results section is very short, and it needs improvement. The authors should discuss and compare the results comprehensively by drawing the necessary tables.
5. The authors need to describe the comparative advantages and innovations of the current topology optimization-based approach for thick shell structures. There are plenty of papers published in this research area.
https://doi.org/10.3390/math10111863
https://doi.org/10.2514/2.1596
The reviewer recommends the authors enhance the literature survey section of the current manuscript and compare the significance of the current approach as compared to the previously published studies.
Reviewer 2 Report
The manuscript presents research related to the optimal design of piezoelectric sensors based on a developed mathematical model. The main advantages of the presented model-based approach are related to accelerating the design process and creating opportunities to obtain sensors with improved characteristics. The manuscript is on a very topical subject and has great potential for development. My main observations and comments are as follows:
- it would be useful, in addition to the advantages in the conclusion section, for the authors to comment on the disadvantages of the proposed model and model-based design approach. In this way, a complete picture of its qualities, capabilities and application limitations will be obtained;
- as many symbols and abbreviations are used in the text, I recommend the authors to add a list of symbols used at the beginning or end of the manuscript or to explain them in the text;
- if the authors have the opportunity, I recommend that an economic analysis and evaluation of the effect of the use of the proposed model-based design approach be made;
- in the conclusion section, the authors comment that, "The results have been compared against different commercial software in order to validate the model", and in the 3 examples considered, there is no comparison with results from other software products.
Round 2
Reviewer 1 Report
accept in the present form
Reviewer 2 Report
The authors revised the manuscript according to my comments and recommendations. I have no further comments.